# Cassava Breeding and Cultivation Challenges in Thailand: Past, Present, and Future Perspectives

**DOI:** 10.3390/plants13141899

**Published:** 2024-07-10

**Authors:** Pasajee Kongsil, Hernan Ceballos, Wanwisa Siriwan, Supachai Vuttipongchaikij, Piya Kittipadakul, Chalermpol Phumichai, Wannasiri Wannarat, Wichai Kositratana, Vichan Vichukit, Ed Sarobol, Chareinsak Rojanaridpiched

**Affiliations:** 1Department of Agronomy, Faculty of Agriculture, Kasetsart University, Bangkok 10900, Thailand; piya.k@ku.th (P.K.); chalermpol.ph@ku.th (C.P.); wannasiri.w@ku.th (W.W.); vichukitvic@gmail.com (V.V.); agred@ku.ac.th (E.S.); agrcsr@yahoo.com (C.R.); 2International Center for Tropical Agriculture (CIAT), Km 17, Recta Cali-Palmira Apartado Aéreo 6713, Cali 763537, Colombia; hernanceballosl54@gmail.com; 3Department of Plant Pathology, Faculty of Agriculture, Kasetsart University, Bangkok 10900, Thailand; wanwisa.si@ku.th; 4Department of Genetics, Faculty of Science, Kasetsart University, Bangkok 10900, Thailand; supachai.v@ku.th; 5Center for Agricultural Biotechnology, Kasetsart University, Kamphaeng Saen Campus, Nakhon Pathom 73140, Thailand; agrwck@ku.ac.th

**Keywords:** cassava breeding, conventional breeding, Kasetsart 50

## Abstract

Cassava (*Manihot esculenta* Crantz) was introduced to Southeast Asia in the 16th–17th centuries and has since flourished as an industrial crop. Since the 1980s, Thailand has emerged as the leading producer and exporter of cassava products. This growth coincided with the initiation of cassava breeding programs in collaboration with the International Center for Tropical Agriculture (CIAT), focusing on root yield and starch production. The success of Thai cassava breeding programs can be attributed to the incorporation of valuable genetic diversity from international germplasm resources to cross with the local landraces, which has become the genetic foundation of many Thai commercial varieties. Effective evaluation under diverse environmental conditions has led to the release of varieties with high yield stability. A notable success is the development of Kasetsart 50. However, extreme climate change poses significant challenges, including abiotic and biotic stresses that threaten cassava root yield and starch content, leading to a potential decline in starch-based industries. Future directions for cassava breeding must include hybrid development, marker-assisted recurrent breeding, and gene editing, along with high-throughput phenotyping and flower induction. These strategies are essential to achieve breeding objectives focused on drought tolerance and disease resistance, especially for CMD and CBSD.

## 1. Introduction: Cassava Domestication and Cultivation Worldwide and in Thailand

Cassava (*Manihot esculenta* Crantz), originally from South America, serves as a primary food crop in numerous countries, notably in Sub-Saharan Africa. Nevertheless, in Southeast Asia, cassava has emerged as a major industrial commodity, particularly in starch-based industries producing modified starch, bio-plastic, and bio-ethanol. Furthermore, it serves a crucial function as a primary source of carbohydrates in animal feed.

The 98 *Manihot* species that have been identified originated in the Americas, ranging from the southern United States to central Argentina [1]. There were two hubs of bio-diversity for *Manihot* species: Brazil for 80 species and Central America for the rest [2]. While many species are endemic in South America (particularly in the Amazon basin region), there is another significant hub of bio-diversity for this genus in Mexico and Central America. The domestication of cassava took place in South America [3,4,5]. The list of *Manihot* species with geographical origin, specific traits and endangered status is available in the Appendix A. Nevertheless, the precise botanical origin of cassava is yet to be determined, and its lineage remains ambiguous. The prevailing hypothesis is that cassava was domesticated once from *M. esculenta* subsp. *flabellifolia* [3,6,7] and the origin of the crop is likely to be the southern Amazon basin (Rondónia and Mato Grosso) [2]. In addition to subspecies *flabellifolia*, there is ample evidence of hybridization between *M. esculenta* and *M. glaziovii* [7,8].

The Portuguese explorers introduced cassava, along with the expertise in manufacturing cassava flour (known as Farinha in Brazil), to Africa in around 1550. During the late 1800s, cassava became a prominent staple crop in Africa. Cassava was introduced to Asia, specifically Sri Lanka, India, the Philippines, Indonesia, and Malaysia. This happened in the 16th and 17th centuries, either directly from the Americas or indirectly through Africa. In the 19th century, the cultivation of cassava became prevalent throughout Southeast Asia [9,10].

Interestingly, cassava production in Thailand has consistently held the top position in Southeast Asia and ranks third globally since 1986 [11]. In 2002, cassava production in Thailand totaled approximately 34 million tons, cultivated across 1,587,369 hectares. The northeastern region accounted for 54.2% of cultivation, followed by the Northern region at 26.0%, and the Central region at 19.8%. The national average yield stood at 21.5 tons per hectare. By 2022, Thailand exported 5,930,697 tons of dried cassava valued at 1485 million USD, alongside 3,631,043 tons of starch valued at 1757 million USD [11]. Thailand has consistently held the top position in the global export market for cassava commodities since the 1960s. It first ranked as the leading exporter of dried cassava roots in 1963 and of cassava starch in 1982 [11]. This achievement reflects increasing demand over time, driven previously by the needs of animal feed production and then shifting to starch-based industries. The country’s robust cassava breeding program in the early 1980s, as noted by Dr. Kazuo Kawano [12], has played a significant role in establishing a centralized national breeding system. This system aims to enhance productivity to meet the demands of both local cassava processing industries and the international markets for dried root chips, starch, and related products.

This review includes the following sections: Section 2. Global Cassava Genetic Resources; Section 3. Special Features of Cassava Physiology, Reproductive Biology, Roots and Starch; Section 4. Cassava Breeding in Thailand; Section 5. Released Commercial Cassava Varieties in Thailand; Section 6. The impact of KU50 in Thailand and Southeast Asia; Section 7. Determinants of Cassava Yield and Production Quality; Section 8. Future of Cassava Breeding Direction.

## 2. Global Cassava Genetic Resources

There are now 13,832 cassava accessions being conserved ex situ in collections located in nine different countries. Colombia, Brazil, and Nigeria have the largest collections, with 5963, 3620, and 3234 accessions, respectively [13]. A comprehensive report on the conservation of *Manihot esculenta* and other *Manihot* species was prepared by Hershey in 2008 [14].

The collection housed at the International Center for Tropical Agriculture (CIAT) in Colombia comprises 5577 accessions of *M. esculenta* and 386 wild relatives from 23 different *Manihot* species. This gene-bank is renowned as the largest repository of cassava genetic diversity. Established initially in the field in 1969, it has been maintained in vitro since 1978 [12,15]. The genetic materials primarily consist of farmer’s landraces and a few elite clones from breeding programs. Approximately 37% of these genetic resources originated from Colombia, 24% from Brazil, 21% from other South American countries, 7% from Central America and the Caribbean, and 7% from Asia. Since 1979, over 43,400 samples from CIAT’s cassava collection have been distributed to 84 countries. Duplicates of the cassava collection held at CIAT are also maintained at the Corporación Colombiana de Investigación Agropecuaria (AGROSAVIA) in La Selva, Colombia, and at the International Potato Center (CIP) in Peru [16].

Since 1975, accessions from CIAT’s cassava collection have undergone evaluation by the cassava breeding programs in many countries importing cassava germplasm, concentrating on productivity, root quality, and responses to both biotic and abiotic stresses across diverse environments [17]. From 2016 to 2021, SNPs and SilicoDArT markers were applied to 5302 cassava accessions for identifying redundant genotypes in the germplasm [15]. A thorough screening of starch quality traits from thousands of accessions has also been published [18].

Significant genetic diversity of both cultivated and wild relatives of cassava is also preserved in Brazil, primarily at the Empresa Brasileira de Pesquisa Agropecuária (EMBRAPA) and the Universidade de Brasília [19]. The EMBRAPA Cassava Germplasm Bank at the Cruz das Almas Experimental Station in Bahia State, Brazil, houses approximately 5000 cassava accessions, which are maintained both in the field and in vitro. These accessions encompass landraces and modern breeding lines from various Brazilian cassava-growing regions and eco-regions, including the Amazon, Cerrado, Pantanal, Caatinga, Atlantic Forest, and Pampa. Genetic diversity studies have identified 1536 accessions as a unique core set of Brazilian cassava germplasm collection, selected through Identity-By-State (IBS) analysis of genotyping data. This germplasm collection originated in the Amazonia region in the north and migrated to the Atlantic Forest in the south and the Caatinga in Eastern Brazil [7].

The germplasm *M. esculenta* collection in Nigeria is hosted by the International Institute of Tropical Agriculture (IITA) and maintains approximately 2712 accessions, comprising mostly landraces from West Africa along with some wild relatives. It is worth noting that elite germplasm and landraces from eastern, southern, and central Africa are currently underrepresented in international collections [20].

In Asia, there are 1700 accessions of cassava collection at the Central Tuber Crops Research Institute of India, 184 landraces at the National Bureau of Plant Genetic Resources, and over 500 accessions of cassava and eight wild *Manihot* species maintained in state universities in India [19].

The cassava germplasm collection for the breeding program in Thailand began in the 1930s with the importation of three varieties from the Philippines and the introduction of 17 varieties from Malaysia to the southern agricultural experiment station. Subsequently, local cassava landraces from northeastern Thailand were gathered at the Rayong Field Crops Research Center, Department of Agriculture (DOA), in 1956. Seven varieties were included in the collection from Indonesia in 1963, followed by the addition of 43 varieties from the Virgin Islands by the Thai Tapioca Trade Association in 1965 [21].

In 1970, Thailand welcomed five varieties (CMC9, CMC36, CMC72, CMC76, and CMC84) from CIAT, which was followed by the introduction of 20 varieties and over 100,000 botanical seeds between 1975 and 1999. From the evaluation and selection of CIAT F1 hybrid seeds, the varieties Rayong 3, Rayong 2, and Rayong 60 were registered and released between 1983 and 1987 (see Appendix A for details on the parentage of Thai varieties). Subsequently, Rayong 3 served as a parent for Rayong 5, which was released in 1994 [22]. Notably, from 2001 to 2005, a memorandum of understanding (MOU) between the DOA, Thailand, and CIAT facilitated the importation of 943 in vitro cassava accessions from the germplasm collection to the Rayong Field Crops Research Center, with the addition of 30 disease-resistant varieties in 2007. In 2013, the cassava breeding program at the Department of Agronomy, Faculty of Agriculture, Kasetsart University, introduced 21 genotypes from CIAT, carrying useful traits such as high-carotenoids content, excellent cooking quality, and resistance to cassava mosaic disease (CMD), bacterial blight, and super-elongation disease [23].

In China, the Chinese Academy of Tropical Agricultural Sciences (CATAS) oversees the national cassava research program and trial network. The cassava breeding program in China was initiated in the 1960s. CATAS maintains a cassava germplasm bank with over 120 accessions, including 30 introduced from CIAT/Colombia and the Thai–CIAT program. In the frost-free conditions of southern China, approximately 3000 hybrid seeds from 80–100 crosses were evaluated, along with 2000–3000 hybrid seeds imported from CIAT/Colombia and the Thai–CIAT program. Among 500 promising lines, OMR33-10-4, ZM8641, and ZM9057 were selected for further evaluation before being released as varieties with high root yield and high dry matter content [24].

## 3. Special Features of Cassava Physiology, Reproductive Biology, Roots and Starch

Cassava is a perennial species with no fixed phenological stage for harvesting. The crop can be harvested from 6 to 7 months after planting. Nevertheless, when exposed to cold or drought stress, it is feasible to harvest it between 18 and 24 months after planting [25]. This remarkable flexibility allows farmers to choose the optimal time for harvesting based on various factors, such as market conditions, environmental outlook, labor and machinery availability, and specific household needs. During the dry season in Thailand, there is an upsurge in the demand for cassava. Therefore, farmers have the opportunity to harvest cassava during this period to take advantage of the increased price, provided they can supply water for cassava production using their own stems for the next crop. Additionally, the crop possesses a remarkable ability to accumulate starch in its storage roots, even under conditions of low soil fertility and water deficit. These combined traits contribute to the exceptional reliability and resilience of cassava, attributes highly valued by farmers and fundamental for ensuring food security.

Cassava can be propagated either from stems or botanical seeds [25]. Stem propagation is the conventional method for commercial production, while seed propagation is essential for breeding purposes. Before harvesting the roots, the main stems are cut and bundled together in groups of approximately 50 stems, each to be stored in an upright position in the shade and on the soil with regularly watering [10]. The length of these stored stems can vary from 1 to 2 m depending on the cultivars having branching or non-branching plant type and growing conditions. The stem will be cut into 20-cm-long stakes with 5 to 7 nodes immediately before planting [10].

Sexual reproduction is common and relatively straightforward to achieve [9,25,26]. Cassava is a diclinous and monoecious species, meaning that either female (pistillate) or male (staminate) flowers are produced in inflorescences (racemes or panicles) within the same plant [25,27]. Self-pollination is feasible except for the limitations imposed by cassava’s protogyny [28,29,30,31,32]. Male and female flowers ready for pollination can only be found on separate branches of plants belonging to the same genotype. In fact, the first sequenced cassava genome comes from an S_3_ partially inbred line [8].

Inflorescences always develop at the apex of the stem. Buds below the inflorescence begin to sprout as soon as flowering is induced, enabling further plant growth [25,33]. Consequently, each flowering event leads to branching. Some genotypes flower early and often (3–5 times during a growth cycle), and others flower infrequently, late, or not at all. Flowering and branching patterns in cassava are highly heritable traits [34]. The inflorescences of the first flowering event generally abort [35]. Consequently, breeders initiate the crosses usually at the second or third flowering event, further delaying the collection of seeds from new progenies.

Farmers often favor erect, non-branching types due to their facilitation of cultural practices and increased production of vegetative planting material, which is easy to transport and store (Figure 1B–D). However, this preference poses a significant challenge for breeders, as genotypes with erect plant architecture tend to flower late and infrequently. Consequently, producing botanical seed from these genotypes may require up to 16 months after planting. In extreme cases, making crosses and obtaining segregating progenies may be practically impossible. Various publications detail the protocols for controlled pollinations in cassava [9,26,33,36]. Achieving synchronization of flowering for planned crosses often presents a significant challenge.

The storage roots of cassava typically consist of about 60–70% water and 30–40% dry matter. The majority of this dry matter content (approximately 85%) in cassava roots is starch, with minimal amounts of proteins and fats. These characteristics contribute to the exceptional quality of cassava starch, which is also relatively easy to extract. Like native starches from other crops, cassava starch exhibits unique functional properties, which make it particularly suited for specific processes and applications [18,37,38,39,40]. Moreover, bio-fortification breeding programs to enhance nutritional quality, have released varieties with higher carotenoid content [41].

Cassava root has particular problems for transport and processing due to physiological post-harvest deterioration (PPD) and cyanogenic content. PPD spoils the roots within a few days after harvest, thus requiring prompt processing or consumption [18,42,43]. PPD is a complex oxidative process involving the reactive oxygen species (ROS) scavenging system resulting in the oxidation of polyphenolics and their glucosides, which are mostly scopoletin and scopolin [44,45,46], and phenotyping it proves to be difficult and time-consuming. Unfortunately, there is a positive correlation between dry matter content (DMC) and PPD [47,48]. On the other hand, a high concentration of carotenoids appears to mitigate PPD [49]. Genetic solution to PPD seems to be difficult and the different value chains must overcome it by proper logistics from the time of harvest through the processing or consumption of the roots.

Cassava leaves and roots contain cyanogenic glucosides [50,51,52], which can often reach toxic levels for human consumption. These glucosides are broken down by the enzyme linamarase, releasing the volatile poison hydrogen cyanide (HCN). While all processing methods effectively release HCN, they may impart a bitter flavor that is often disliked by consumers.

## 4. Cassava Breeding in Thailand

Cassava breeding was established around the 1930s in East Africa by the British in Tanzania, and in Brazil [53]. Two international research centers working on cassava breeding were created in 1967. The International Center for Tropical Agriculture (Centro Internacional de Agricultura Tropical, or CIAT) was founded in Palmira, Colombia and the International Institute of Tropical Agriculture (IITA) is based in Ibadan, Nigeria. Thereafter, several National Research Programs conducting cassava breeding were established in many countries in Asia, Africa, South America and the Caribbean. In the early 1980s, CIAT initiated a productive and long-standing collaboration with DOA, Thailand [12].

The main stages in a cassava breeding program are as follows: 1. germplasm collection, either from local landraces or from introduction of accessions from other countries; 2. germplasm evaluation for adaptation and parent selection for crossing; 3. crossing to generate breeding populations; and (4) evaluation and selection in field evaluation trials. These trials can be grouped as: (4.1) seedling selection trial from botanical seeds; (4.2) clonal selection trials or single row trials (SRTs); (4.3) preliminary yield trials (PYTs); (4.4) advanced yield trials (AYTs): (4.5) regional yield trials (RYTs), which involve many years, seasons and/or environments; and (4.6) registering and releasing of variety [23].

Kawano [54] proposed that a crucial factor in selecting parental material for cassava breeding programs is their ability to yield a high number of seeds. This is because the segregating families would exhibit heterogeneity, necessitating selection at the seedling trial stage. Therefore, a larger quantity of seeds increases the likelihood of identifying desirable genotypes. At the cassava breeding program in CIAT, approximately 40,000 to 60,000 botanical seeds are produced annually from 200 to 400 crosses, with a survival rate of around 75%. From the seedling nursery, 500 to 2500 genotypes (1.7–5.6%) advanced to the Single Row Trial (SRT). Subsequently, 100 to 300 lines (12–20%) were evaluated in the Preliminary Yield Trial (PYT), and 40 to 80 lines (26–40%) progressed to the Advanced Yield Trial (AYT). Promising lines in the multi-location Regional Yield Trial (RYT) could number between 10 and 30 (25–37.5%).

As selection in conventional breeding was based on phenotypic evaluation, it is suggested that, in the early selection trials, estimation of phenotypic breeding values and multi-location trials should be performed at the SRT stage, with the assistance of rapid stem multiplication. Cassava breeding relies on phenotypic recurrent selection within highly variable full-sib (from directed crossing) or half-sib (from open-pollination) families. The major obstacle in cassava breeding is the significant genetic variability observed among F_1_ progenies, even within full-sib families. This within-family genetic variability, predominantly found in crosses derived from heterozygous genotypes, constrains the selection of progenitors based on general combining ability or breeding value [29,55], especially for complex traits, such as fresh root yield (FRY) [56]. Ongoing efforts are dedicated to developing models aimed at mitigating the adverse effects of within-family variation on the predictive capacity of genomic estimated breeding values [57]. Another significant consequence of employing heterozygous progenitors is the inability to utilize the back-cross scheme for trait introgression. Difficulties in introgressing single-gene traits, such as waxy starch [58], or resistance to Cassava Mosaic Disease (CMD) serve as clear examples of this bottleneck.

Genotype-by-environment interactions play a significant role in shaping complex traits across various crops, including cassava. Addressing these interactions requires thorough multi-location testing, especially for traits like FRY and other low-heritability quantitative characteristics. A minimum of 8–12 months is required for harvesting roots and evaluating starch content. Moreover, the multiplication rate of cassava stems is relatively slow; typically, a single plant from botanical seed yields 6–8 clonally replicated stems (subject to variation based on seedling phenotype) for subsequent SRT. Consequently, the entire breeding cycle, spanning from crossing nurseries to the selection of clones for potential release as varieties, extends over 6–8 years.

In the first year of the breeding program, the crossing nursery is designed attending to the breeding objective(s) and a varying number of accessions are selected as parents. A large number of F_1_ hybrid seeds should be obtained from several crosses, as in the example from the CIAT cassava breeding program mentioned earlier. All seeds were germinated in seedling pots for 45 days before transplanting in the seedling nursery [12].

In the second year, The F_1_ seedling trial must be highly consistent because each plant has a unique genotype, which is distinctive from the others. Selection of single plants should be made carefully because of the relatively high possibility of eliminating desirable genotypes (because of adverse environmental conditions) or selecting poor ones (because of particularly favorable growing conditions). Therefore, the seedling trial is regarded as a seedling nursery instead and the main objective is to grow healthy plant materials for the following SRT. Some cassava plant characteristics with high-heritability can be selected in the seedling nursery, such as cassava mosaic disease (CMD) resistance in high disease pressure areas in Africa. In Thailand, selection is done for plant type (erect architecture), adequate vigor and acceptable harvest index (HI) and, occasionally, for special characteristics such as waxy starch [58].

In the third year, the SRT typically serves as the initial clonal evaluation, often conducted in a single location. However, if planting materials are sufficient for the selected lines to proceed to the PYT, Kawano [12] recommends conducting two additional SRTs in different locations. A notable modification to the standard breeding pipeline has been implemented at CIAT in Colombia. The seedling nursery is grown off-season for only six months. At the harvest, selection for high heritability traits, such as plant architecture, harvest index, special starch properties, pigmentation in the parenchyma, and/or disease resistance, can be conducted. Since the plants are young, only three stem cuttings can be collected. These cuttings are then planted at the normal planting time in a stage referred to as F_1_C_1_. The resulting three plants are grown for a year, akin to conventional seedling nurseries. The key distinction lies in having three plants per genotype, allowing for the subsequent planting of three distinct single row trials in three different locations the following season [53]. However, due to Thailand’s climate, this new system requires slight adjustments. Off season planting would start in November, which is in the middle of the dry season in Thailand, and thus would be exposed to water deficit stress during vegetative growth. This new system, therefore, is best suited for well-irrigated areas.

Kawano [54] recommended against selecting genotypes in the early stages of the selection process (e.g., seedling nursery and SRT) based solely on FRY or biomass, but rather suggested using HI values for selection, as this demonstrated a stronger positive correlation with FRY in later evaluation stages compared to using FRY values at the early stages as the selection criteria. HI is the ratio between root biomass and total biomass.

Each row in the SRT typically comprises 6–8 clonally propagated plants from the same genotype. Normally, there are 500 to 2500 genotypes in the SRT, where FRY, starch content (SC), and other harvesting traits are typically assessed. In CIAT/Thailand collaborative breeding programs, the spacing within the row was typically 1 m apart, while the spacing between rows was 2 m apart. This configuration allows for ample space for evaluating root bulking potential, reduces soil nutrient competition, and prevents shading from neighboring rows with taller plant types. This spacing approach was also employed in the seedling nursery [12].

From the SRT, 100–300 genotypes (approximately 10–20% of all genotypes in the SRT) are selected. For yield potential evaluation, cassava plants were harvested at 11.5 months after planting (MAP) for FRY, total biomass and root dry matter (e.g., SC) and compared to those from three check varieties in the initial of CIAT/Thailand collaborative cassava breeding programs [12]. However, this extensive selection at only one location can eliminate some genotypes which may respond well in the other environment due to genotype-by-environment (GxE) interaction. This is a problem partially overcome by replicating each SRT in three different locations as mentioned above.

In the fourth year, the PYT consists of 100–300 genotypes, with 20 plants per plot per location. In Thailand, each genotype can be grown in one location with two replications. If SRT were conducted in multiple locations, the PYT could be expanded in plot size across the same number of locations as in the initial SRT. In each plot, there may be up to 35 plants within a 5 × 7 square meter area, of which 3 × 5 plants can be harvested for yield evaluation, while the border plants remain untouched. However, a second modification introduced at CIAT [53] involves planting PYTs in plots with 10 plants and 3 replications across 3 locations. For speed breeding, the use of 8-cm planting materials was investigated in place of 16-cm conventional planting materials. With this rapid cassava propagation method, PYT could be skipped and the breeding period could be reduced from 11 years to 9 years [59].

In the fifth year, the AYT comprises 40–80 genotypes, representing approximately 25–40% of all genotypes in the PYT. These are planted with three replications in plots containing 25 plants per genotype. Subsequently, RYT are grown in multi-environmental locations with 10–30 genotypes. Variation in soil type and climate constitutes crucial factors in these multi-environmental trials. RYTs are typically conducted in both the wet (early rainy) and the dry (late rainy) seasons, encompassing a range of conditions from fertile soils with well-distributed rainfall to challenging environments, such as acidic soil, low fertility soil, high disease pressure, and/or prolonged drought areas [12].

The collaborative cassava breeding program between CIAT and DOA, Thailand, operated from 1982 to 1998. The Thai–CIAT breeding program produced between 20,000 and 40,000 F_1_ seeds annually. Approximately 12,000 to 25,000 genotypes were then evaluated in the seedling trial at the Rayong Field Crops Research Center, from which 1200 to 2400 genotypes advanced to the SRT. Subsequently, around 100 to 160 clones were selected and grown in PYT with two replications. The initial three cycles of selection, spanning from the seedling trial to the PYT, took place at the Rayong Field Crops Research Center. Following this, approximately 16 to 20 clones were chosen for evaluation in four replications in AYT across three locations. Furthermore, seven to nine promising clones were selected for evaluation in regional selection trials across six locations [12].

The Kasetsart University Cassava Breeding Program, situated at the Sriracha Research Station, was launched in 1984 through crosses between 93 promising lines and Rayong 1 (R1), a highly favored variety in Thailand at the time. R1 stood out for its exceptional traits, including a high germination rate even in dry seasons, an upright and late branching plant architecture, resilience to local diseases and pests, and commendable adaptability, despite its moderate root yield and starch content.

Breeding objectives focused on achieving high HI and total biomass, including a large plant type, numerous roots, and high root starch content. From the 93 crosses, a total of 11,151 seeds were obtained, resulting in 7884 plants grown and evaluated in the seedling nursery. At 11 MAP, F_1_ plants were harvested and assessed for erect plant type, late branching, root size, HI, and SC. Following seedling selection, 435 lines proceeded to the SRT the following year. In the SRT, ten plants per genotype were compared with R1 as the check variety, with evaluation criteria including germination rate, plant type, FRY, SC, and HI. FRY was reported to have direct effect for SC. However, selecting only FRY cannot successfully guarantee high SC [60]. Therefore, both traits have become the major criteria for selection. Among the 435 lines in the SRT, 79 clones were selected for the PYT, comprising eight plants per plot and two replications. Among these 79 clones at the Sriracha research station, 27 were further chosen based on criteria such as germination rate, plant type, and SC for evaluation in the AYT, with Kasetsart 50 (KU50) being one of the selected lines. From 1987 to 1991, a total of 60 RYTs were conducted across 15 provinces in eastern, northeastern, and central Thailand through collaboration between Kasetsart University, the Rayong Field Crops Research Center of the Department of Agriculture, and with the assistance of Dr. Kazuo Kawano, Regional Cassava Program from CIAT [61].

## 5. Released Commercial Cassava Varieties in Thailand

The success of the Thai–CIAT collaborative cassava breeding program, along with collaborations with other countries in Southeast Asia, can be attributed to a combination of factors. Firstly, the introduction of valuable genetic diversity from international germplasm resources played a pivotal role. Additionally, effective evaluation in diverse environmental conditions, spanning from highly fertile to less fertile locations, contributed significantly to the program’s success [12]. The close collaboration between research and industry institutions, as Thailand has been one of the leading countries exporting dry cassava and starch globally, coupled with favorable environmental conditions and the absence of major pests and diseases, undoubtedly also played significant roles in this success story.

Cassava breeding programs in Thailand have focused on starch industry end-uses. Thus, the main characteristics for evaluation are FRY, SC, HI, leaf and stem weight, total plant weight and erect plant type. Starch quality traits, such as white root flesh and high viscosity, are also favored by breeders. Other agronomic traits, such as high germination rate and delayed post-harvest physiological deterioration, are also relevant [62].

The cassava breeding program in Thailand from 1957 to 2015 has been partitioned into four periods starting with the release of R1, as follows [23];

1. From 1957 to 1974: There was no registered variety in Thailand and only landraces were grown by farmers. The average FRY of cassava was only 14.8 t/ha;

2. From 1975 to 1982: R1, a landrace variety, was registered as the first Thai commercial variety by the DOA, Thailand. Although this variety was officially promoted, the average yield of cassava remained the same as during the first period (14.9 t/ha);

3. From 1983 to 1991: During this period, five improved varieties were released. Rayong 2, Rayong 3, Rayong 60 and Rayong 90 (R2, R3, R60, and R90, respectively) were developed by DOA. The fifth variety, Sri Racha 1 (SRC1), was released by Kasetsart University. However, the average yield of cassava in Thailand still remained low (13.7 t/ha);

4. From 1992 to present: this was a flourishing period for cassava in Thailand. Several modern varieties, with outstanding performance, were released and successfully adopted by Thai farmers. Consequently, yield of cassava in Thailand increased to averages ranging between 17.7 and 22.0 t/ha [23], whereas the yield of these varieties in the research yield trial were reported ranging from 27.5 to 41.3 t/ha [21]. As demonstrated in Figure 2, the slow growth cassava production in Thailand compared with those from Southeast Asia was due to the fact that the country’s average yield had reached the plateau, while the optimum root yield from the newly released varieties was still high.

The primary objective of cassava breeding in Thailand is to maximize root yield and starch content, crucial for supporting various industrial sectors. Markets for products with additional quality traits, such as high carotenoid content or superior cooking qualities, remain somewhat limited. However, the price of cooking cassava, particularly the Hanatee variety grown in irrigated areas, commands more than five times that of industrial cassava. Varieties released by the Rayong Field Crop Research Center are assigned unique codes for registration purposes: odd numbers for industrial varieties and even numbers for cooking varieties. Among these, Rayong 2 is the only variety with an even code (excluding R60, R72, R90, which were named according to specific celebratory years), distinguishing it as the sole yellow-fleshed cultivar specifically developed for cooking purposes.

## 6. The Impact of KU50 in Thailand and Southeast Asia

KU50 stands out as the most renowned cassava variety in Thailand and Southeast Asia, recognized as KM94 in Vietnam, NSIC cv 22 in the Philippines, and Ai-Luka 1 in Timor-Leste [61]. Released in 1992, KU50 emerged from a collaborative cassava breeding program involving three organizations: Kasetsart University, the Department of Agriculture of Thailand, and CIAT. Since its release, KU50 has garnered favor among farmers and starch processors across Thailand and Southeast Asia, swiftly spreading throughout the region. Its popularity stems from its consistent high root yield across diverse environments, coupled with a robust sprouting rate and survival rate (approximately 94%) even under water stress conditions. KU50 has relatively high root dry matter and starch content, further enhancing its appeal and widespread adoption.

KU50 originated from a cross made in 1984 at the Sriracha Research Station in eastern Thailand between R1 and R90, serving as the female and male progenitors, respectively. R1, released in 1975, is a landrace with an unknown origin. On the other hand, R90, released in 1991, was derived from a cross between CMC76, a Colombian landrace, and V43 from the Virgin Islands. While R90 exhibits higher FRY and SC compared to R1, it has a lower sprouting rate and tends to branch earlier. Additionally, R90 demonstrates lower FRY stability when compared to R1. Praneetvatakul et al. [63] evaluated the benefits of research investment of KU50 and the net present value of research from the released year until 2014 on KU50 was about 1.32 billion US dollars.

R1 serves as the ancestor of numerous Thai varieties, including R60, SRC1, KU50, R72, HB60, HB80, R7, and R84-13 (full variety name is shown in Appendix A). Kittipadakul [23] highlighted that approximately 56% of the Thai commercial genetic background of cassava can be traced back to R1. Similarly, Rayong 5 emerges as the ancestor of several Thai varieties, such as R72, R9, HB60, HB80, R11, KU72, and R84-13. Given their shared ancestry, inter-varietal crossing utilizing Thai commercial varieties may have limited potential to surpass the yield capabilities of KU50. However, many Thai commercial varieties became promising progenitors for the cassava breeding program in CIAT when crossed with varieties with diverse genetic background. In a large study including thousands of progenies evaluated through 14 years of SRT, the progeny from R90 showed the highest average selection index [19].

Later in 1994, R5, from a cross between 27-77-10 and R3, was released by DOA with characteristics of high HI and high total biomass yield. Since R5 and KU50 are genetically unrelated based on the available pedigree information (Figure 3), the cassava breeding program at Kasetsart University focused on a cross between them. As a result, Huaybong 60 (HB60) and Huaybong 80 (HB80), having high FRY, HI, SC and total biomass yield, were released in 2003 and 2008, respectively. These two varieties were siblings from a cross between KU50 and R5 [23]. In addition, HB90 and HB100, released in 2017 and 2022, were also from the open-pollinated MKUC34-114-235 line, which was derived from the same cross as HB60 and HB80, as shown in Figure 3 [64].

## 7. Determinants of Cassava Yield and Production Quality

### 7.1. Yield Stability

Different varieties exhibit varying performance across different environments due to genotype by environment interactions (GxE). Environmental factors, such as locations, seasons, and years, contribute to the differentiation of these environments, leading to variations in rainfall, temperature, insect and disease pressure, and soil fertility. These environmental factors significantly influence cassava yield parameters, including FRY, dry root yield (DRY), SC, and HI [23].

Cassava FRY is prone to instability, particularly due to variations in soil moisture. Achieving high and stable SC is a highly desirable objective. This trait exhibits a low positive correlation with FRY in the early stages of selection [54]. However, research has shown that this initially weak and positive correlation gradually transforms into a negative and robust relationship in successive evaluation stages [65]. The selection process predominantly favors those with high DRY, which essentially represents the total amount of starch accumulated in the roots. Genotypes are favored in the selection process if they prioritize high FRY, high DMC, or achieve compromising levels for both variables simultaneously. It remains challenging for a genotype to reach maximum levels for both FRY and DMC concurrently, as doing so would require a substantial demand for starch, which is a limited resource for the plant. Jennings and Hershey [66] indicated that maintaining high DMC while continuing to increase FRY would be difficult.

High and stable SC offers greater reliability across various evaluation stages, making it highly prized by both farmers, who seek higher root prices, and industrial processors aiming for improved production efficiency. Varieties that prioritize FRY at the expense of mediocre SC levels often face rejection by the industry, as exemplified by the case of R60, which boasted high DRY but had unacceptable SC levels [62]. Environmental stress, such as drought, can trigger starch mobilization from the roots for bud regeneration upon rainfall. Therefore, the focus on SC characteristics revolves around minimizing starch reduction after periods of drought [67].

Broad-sense heritabilities of DRY, FRY, and biomass are typically low at the early stages of evaluation [54] but improve with larger replicated plots in later stages, which also include multi-location trials. Therefore, AYT and RYT in multi-locations are important for evaluation and intensive selection of cassava breeding genotypes to reduce the negative impact of GxE.

Stability of performance is critical for the success of a release variety. There are different approaches to assess stability. The regression of FRY of individual genotypes on the location mean (across many genotypes) in several contrasting environments [68] was used to assess the performance and stability of elite Thai varieties [23,62]. The regression coefficient is a useful indicator of stability. Stable varieties have regression coefficients around 1.0. Genotypes with b < 1.0 are particularly adapted to low-yielding environments while those with b > 1.0 perform well only when growing conditions are favorable. In the case of R1, b = 0.84 (for FRY) indicates a tendency to have better performance in unfavorable conditions. Rayong 60 tends to perform particularly well in favorable environments (b = 1.07). R3, R90 and KU50 showed desirable stability for FRY (b = 1.02; 0.98 and 0.96, respectively). KU50 showed the highest dry root yield, which was explained by outstanding FRY and acceptable levels of DMC (Table 1).

Compared with the three later-developed varieties (HB60, R5 and R72), KU50 still had relatively high stability with regression coefficient of starch content, fresh root yield, and dry root yield in 11 location trials in the northeastern and eastern Thailand (Table 2) [69].

### 7.2. Environmental Factors Determining Cassava Yield and Starch Production

Howeler [70] identified environmental factors, including soil conditions, that influence cassava productivity across Africa, America, and Asia. Various elements affect the performance of cassava varieties, including weather patterns, plant diseases, pests, soil fertility, erosion, salinity, and surface soil temperature. Effective soil fertility management can enhance cassava productivity by 27% in Africa and the Americas, and by 32% in Asia. Regarding planting management, several considerations are vital in plot preparation, including the use of high-quality cuttings, appropriate spacing between plants, and implementing effective weed control measures [70]. Properly utilizing quality cuttings and employing efficient weed management strategies can elevate productivity by 17–23%. Incorporating high-yielding cassava cultivars can further increase the landscape’s output potential by 26–29%. Extreme weather events, such as droughts and flooding, significantly affect cassava output. Reducing water stress can elevate cassava output by 13–25%, while managing waterlogging can boost cassava yield by 28–53%. Additionally, effective management of diseases and pests can result in a 10–20% improvement in cassava productivity [70].

The information provided above offers valuable insights for breeders in identifying potential breeding objectives. While active breeding for low fertility conditions has not been extensively conducted, the evaluation process typically occurs in low fertility conditions, often within farmers’ fields and in accordance with prevailing cultural practices. Breeders consistently prioritize selecting genotypes that produce high-quality planting material. Opting for an erect plant architecture leads to the production of long stems that are easily manageable, thereby reducing the risk of physical damage during transportation and enabling longer storage periods without dehydration. The prompt, robust, and dependable sprouting of cuttings shortly after planting is an important trait that breeders carefully consider during the selection process.

Genetic transformation and gene editing research have focused on herbicide tolerance in cassava, among several other traits. However, regulatory constraints have hindered the release of commercial varieties using these approaches. Therefore, a unique opportunity and a pressing need exist to screen cassava genetic resources for natural sources of tolerance to herbicides, particularly those interacting with Aceto-hydroxy-acid synthase (AHAS) and acetolactate synthase (ALS). Natural tolerance to such herbicides has been observed in various crops [71,72]. ‘Molecular sieving’ techniques, like Eco-Tilling, offer promising alternatives, and initial strides have been made in this direction [73].

Cassava has a recognized tolerance to water stress [74]. Being a perennial plant, it has the flexibility to stop growth when conditions are not favorable and to resume it when they improve. It has a remarkable stomatal sensitivity, which regulates evapotranspiration under water stress [75]. There are several reports demonstrating large genetic variation for tolerance to drought in cassava [76,77,78,79,80].

However, drought drastically affected cassava growth and yield as root yield was 72.98%, shoot yield was 54.95%, dry matter content was 26.15%, harvest index was 31.05% and plant height was 32.95% from the capacity under irrigated field conditions [79]. Moreover, selecting for drought tolerance in cassava poses several challenges. Firstly, defining water stress for cassava is not straightforward. Typically, cassava experiences severe water stress towards the end of its normal growing cycle due to the common absence of rainfall 2–4 months before harvest. Additionally, there is no proxy trait readily available for selection purposes. For instance, in maize breeding for drought tolerance, a short anthesis-silking interval has been effective because it exhibits higher heritability than yield and is strongly correlated with it under water stress conditions [81]. However, in cassava, the selection for drought tolerance only did not associate with high root yield under drought condition [79].

Carbon isotope discrimination [75], abscisic acid [76,77] and the root length-to-girth ratio [78] have been linked to cassava’s response to water stress. Other various traits have also been associated with ‘drought tolerance’ in cassava [82,83,84,85]. However, none have been utilized in the selection process thus far.

Further research, therefore, is urgently needed to develop efficient alternatives for the phenotypic selection of drought tolerance in cassava. Despite challenges in defining drought tolerance in cassava and assessing it phenotypically, several genomic studies have been undertaken [80,86,87,88].

For varieties in Thailand, Rayong 72 (R72), grown in the early rainy season (in May), was reported to be drought tolerance during a 4-month drought period in the mid growth stage, as it showed the highest leaf retention, which resulted in the highest storage root yield [89]. However, in northeastern Thailand, farmers prefer to grow cassava after rice cultivation, which is typically in the late rainy season. The yield comparison of cassava varieties grown after rain-fed lowland rice cultivation showed that Rayong 7 (R7) provides the highest FRY (in t/ha) at 28.2 followed by R72 (21.3), KU50 (23.2), R11 (21.3) and HB80 (19.2). However, considering starch yield, the highest starch yield (in t/ha) was in KU50 (6.0), followed by R7 (5.5), R72 (5.3), R11 (5.0) and HB80 (4.9) even though R11 provided the highest leaf yield [90].

KU50 was classified as a drought tolerant variety due to its water use efficiency and high biomass in the highland northeastern part of Thailand [91]. For the late rainy season, if irrigation is available during the drought period at the early growing stages, the yield and starch of cassava would be considerably higher than those without irrigation. Polthanee and Srisutham [92] demonstrated that HB80 receiving 30 mm of irrigated water for every cumulative pan evaporation value of 40 mm yielded 92.5 t/ha of FRY, while, without irrigation, FRY was around 51.1 t/ha.

There was also evidence that cassava root yield could be improved under an irrigated environment. Kittipadakul et al. (2017) [23] reported the FRY of the breeding line CMR 34-08-89 (CMR89) of 67.5 t/ha with SC of 18.7% and the variety Huaybong 60 (HB60) with FRY of 51.9 t/ha and SC of 30.2%. The application of a crop simulation model for different growing periods from January to December suggested that growing cassava in February would provide the highest root yield in all varieties [93].

There is not much information on the reaction of cassava to water logging or soil salinity, nor have any breeding efforts attending these limiting factors been reported.

### 7.3. Biotic Factors Determining Cassava Yield and Starch Production

Cassava mosaic disease (CMD) stands out as the most devastating pathogen impacting cassava production across African and Asian continents [94,95,96]. Remarkably, Southeast Asian countries, including Thailand, had previously benefited from the absence of this disease until its recent introduction [97,98]. Cambodia was the first country in Southeast Asia where the disease was initially reported [99], and from there it swiftly spread to neighboring countries.

The outbreak of CMD in the Sisaket and Surin provinces of northeast Thailand was first reported in 2018 by the Department of Agriculture (DOA) [100]. A preliminary CMD survey conducted in Thailand from 2018 to 2019 identified the causative agent as the Sri Lankan cassava mosaic virus (SLCMV), identical to the strain detected in Cambodia and Vietnam [97,99]. The disease gradually spread, affecting vast areas planted with cassava across 21 provinces, with Nakhon Ratchasima, the largest cassava production region in Thailand, being particularly impacted. Projections suggest that, if the CMD outbreaks in Thailand remain uncontrolled, losses valued at 3412 million US dollars could occur from 2021 to 2031. According to research conducted by Witsanu Attavanich [101] in 2022, the infected area is anticipated to reach 532,850 hectares out of a total cassava plantation area of 1,725,188 hectares, with an estimated damage value of 356 million US dollars in 2025 alone.

The different virus strains inducing CMD are members of the Begomovirus family. The virus is transmitted thought infected cutting stem and whitefly vectors, such as *Bemisia tabaci* [102]. According to Uke et al. [103], the planting of infected cuttings in Vietnam led to a reported decreases in cassava root yield by 16–33% and starch content by 22–28%. The alternative hosts of SLCMV are Euphorbiaceae species, including *Cnidoscolus chayamansa*, *Jatropha curcas* and *J. multifida*, as reported in Thailand. The symptoms of SLCMV infected primary and alternative hosts are similar, displaying mosaic patterns, yellowing, and leaf curling [104,105].

The most effective approach to the control of CMD is through the utilization of resistant cassava varieties, a common strategy for plant virus management. A stable and reliable source of resistance (*CMD2*), conferred by a single major gene located on chromosome 12, was identified and mapped in the 1990s [106]. Furthermore, two additional loci on chromosome 14 have been identified [107]. Molecular markers for *CMD2* are readily available and have been extensively employed for selection purposes in both Africa and, recently, Southeast Asia [108,109]. Thailand implemented preventive measures and introduced sources of resistance as early as 2013 (as indicated above). Consequently, well before the introduction of the disease, the introgression of *CMD2* into Thai elite germplasm had already commenced.

In 2018, an additional five CMD-resistant varieties (TMS-IBA980505, TMS-IBA972205, TMS-IBA 920057, TMS-IBA 980581, and TMEB 419) were imported to Thailand from the International Institute of Tropical Agriculture (IITA), Nigeria, through collaboration between Kasetsart University and the Thai Tapioca Development Institute. These varieties were scheduled for release to starch companies and farmers facing high disease pressure in 2023. Further assessment of cassava germplasm cultivated in Southeast Asia [109] revealed sources of resistance distinct from *CMD2*, associated with genotypes CR63 (known in Thailand as Hanatee) and VNM8 (known in Vietnam as Xanh Vinh Phu). Interestingly, KU50 also frequently exhibits moderate levels of resistance.

Cassava witches’ broom disease (CWBD) was initially reported in 2008 in Rayong province and subsequently appeared in Kampangpet and Nakorn Rachsima provinces [110]. Presently, CWBD has emerged as a significant issue in cassava production in Thailand, particularly affecting the cassava production areas in the eastern part of the country.

The symptoms of CWBD include phyllody leaves in the middle and top sections of the plant, along with a reduction in internodal length, resulting in decreased root production. From 2021 to 2023, symptoms of witches’ broom disease primarily manifested when the cassava plant was 9–10 months old or just before the harvesting period in Thailand.

Klinkong [111] identified phytoplasma as the causative agent of CWBD. The presence of phytoplasma in the phloem was observed through electron microscopy, and the Koch postulate was utilized to confirm disease transmission. However, Leiva et al. [112] recently reported CWBD symptoms in cassava plants in Cambodia, Vietnam, and Lao PDR caused by the Ceratobasidium genus. This marks the first documented occurrence of CWBD associated with *Ceratobasidium* sp. in Southeast Asia.

Confirmation of *Ceratobasidium* sp. as the causative agent of CWBD in Thailand has not yet been established. This lack of confirmation underscores the necessity for further investigation to identify and ascertain the actual causal agent of CWBD. Understanding the disease’s etiology is crucial for the development of effective management and control strategies in the future. Fortunately, there is empirical evidence suggesting that Rayong 11 has, for unknown reasons, promising levels of resistance to the disease [113].

A third biotic stress severely impacting cassava productivity in Thailand arose from the introduction of the mealybug *Phenacoccus manihoti* Matile-Ferrero in late 2008. Upon its arrival in Asia, it spread to its ecological limits, resulting in a 27% decline in the nation’s aggregate cassava FRY (from 22.7 to 18.6 ton/ha) and a staggering 162% increase in starch prices [114]. Resistance to this mealybug was not reliably found, prompting the adoption of a suggested strategy: biological control, achieved through the introduction of the *Anagyrus lopezi* De Santis parasitoid wasp [115]. This approach proved highly successful, with the parasitoid rapidly dispersing after the initial mealybug infestation. Consequently, cassava productivity rebounded to normal levels within a few years [114]. In this scenario, breeding for resistance to the insect was deemed unnecessary.

### 7.4. Starch Quality Traits

A waxy starch mutant was identified from germplasm screening in 2006 by Ceballos et al. [116]. From this mutant, a research agreement was signed for development of waxy commercial varieties in Thailand. The first batch of waxy-starch cassava varieties were released in Thailand in 2013 [117]. Due to the recessive trait, way cassava lines will be obtained in F_2_ population as the cross between the waxy starch mutant, AM206-5, and Thai commercial varieties generate only heterozygous in the waxy locus (WXwx). However, with the single-nucleotide amplified polymorphism (SNAP) markers, F_1_ lines with the waxy allele (wx) were selected precisely for generating an F_2_ population [58]. Even though the root yield of improved lines of Thai commercial waxy lines showed similar root yield as non-waxy commercial varieties, the starch content and harvest index were still lower than those of non-waxy lines, suggesting the linkage between waxy locus and the unfavorable locus controlling starch and harvest index properties [56].

Furthermore, a small-granule starch was obtained from gamma-ray mutagenesis of botanical seeds [118], which could be used as the genetic source for developing specialized starch properties in the cassava breeding program in Thailand. This has the potential to be utilized for texture modification and as a substitute for fat in the future.

## 8. Future of Cassava Breeding Direction

Cassava breeders often face limited funding due to their low priority in agricultural policy and the absence of private sector investment in cassava breeding programs. The collaborative operation among stakeholders is mandatory to continue cassava varietal development in order to serve the requirement of all sectors in the value chain, from farmers to the marketplaces and industries.

Private and public breeding programs procure their financial resources and set their priorities in contrasting ways. Public research institutions are not as flexible and amenable to fostering synergies as compared to the private sector [119]. Moreover, public programs are typically the only ones targeting non-profitable markets, such as cassava. Therefore, cassava breeding programs in Thailand and worldwide are essentially public in nature. Decision-making on resource allocation in the public sector remains a critical, and sometimes vulnerable, process. There is an increased risk of investments being technology-driven (e.g., developing and applying cutting-edge technologies), rather than having outcome-driven research agendas focused on solving priority problems (e.g., developing outstanding cassava varieties). Cassava breeding everywhere requires the most appropriate methods and tools to be selected wisely. Marker-assisted selection (MAS) and genomic selection, for example, should not be an end in themselves but tools to make cassava breeding more efficient.

### 8.1. Efficient Exploitation of Heterosis and Non-Additive Genetic Variation

Cross-pollinated crops, such as maize and cassava, demonstrate significant heterotic effects for many relevant traits, particularly yield. Dominance, over-dominance, and epistasis are key components of heterosis. Interestingly, inbreeding depression is a common phenomenon associated with crops and traits where heterosis also plays a crucial role. Moreover, it has been suggested that heterosis and inbreeding depression are, to some extent, opposite sides of the same phenomenon [120]. In other words, the same factors that increase performance in the heterozygous hybrid are related to inbreeding depression in the homozygous progenitors.

General (GCA) and specific (SCA) combining ability represent the additive and non-additive genetic variation, respectively. These parameters are typically quantified through tailored genetic designs, such as test crosses, North Carolina II, or diallel mating designs. Several reports indicate a high SCA/GCA ratio on relevant traits in cassava, such as FRY in Thailand [121] and elsewhere [34,122,123,124,125]. Relevant non-additive genetic effects have also been detected for FRY through molecular studies [35,126,127].

Following Miranda Filho’s [120] suggestion, it is expected that, if heterosis is important for FRY, so too should be inbreeding depression. Similarly, for traits depending mostly on additive genetic effects, such as DMC, by default, heterosis is weak, and so should be inbreeding depression. These expectations have been demonstrated in field experiments [29,31,34]. Indeed, heterosis and inbreeding depression are ‘two sides of the same coin’.

The two predominant methods for exploiting hybrid vigor are Reciprocal Recurrent Selection (RRS) and inbred line improvement (ILI), each originating from distinct heterotic groups. RRS has found application across various crops [128], while ILI stands as the cornerstone for maintaining remarkable and consistent genetic gains in maize, dating back to the inception of double-cross hybrids nearly a century ago [129]. Despite their efficacy, both strategies hinge on the identification (or creation) of heterotic groups, an aspect largely overlooked in cassava.

The challenge in identifying heterotic groups is primarily attributed to the substantial genetic variation within families resulting from the utilization of heterozygous progenitors. This problem is not unique to cassava but extends to other crops, like maize. Unlike maize breeders, who typically evaluate entire full-sib families as a genetic entity without considering individual genotypes within the family, cassava breeders tend to focus on individual genotypes, neglecting the significance of families. The cassava breeding system inadequately addresses the crucial assessment of GCA and SCA, which are essential for the identification and exploitation of heterosis. Previous efforts to transition cassava breeding towards identifying GCA values were abandoned due to inherent complexities mainly attributed to the within-family genetic variation [55]. This problem, incidentally, also affects alternative breeding approaches, such as genomic selection, because of differences in the standard deviation of gamete breeding values (SDGBV) between progenitors at the haplotype level [130].

Kittipadakul et al. [131] assessed GCA and SCA using the range and average percentage of progeny (in seedling selection trial) selected from female and male parents relative to all breeding lines developed from the parents. To determine SCA, a wide variation in the percentage of progeny selected from all parent lines, whether as male or female progenitors, indicates that a parent exhibits strong SCA in a particular cross. For example, HB60, a female parent, produced progeny with a selection range of 0.6–30.9%, meaning that, when HB60 was used as the female parent in crosses, the likelihood of selecting progeny may be as low as 0.6%. When crossed with another male parent, the progeny had a 30.9% probability of being chosen, demonstrating that this cross resulted in high SCA.

Interestingly, it is obvious that R1 and R90 harbor the requisite alleles to produce exceptional hybrids like KU50. In other words, the progenitors of KU50 can be postulated as proven representatives of two potential heterotic groups. One viable avenue to exploit this ‘genetic capital’ could involve a reverse genetic approach to develop (partially) inbred cassava lines from R1 and R90, respectively, resembling to a large extent the gametes that originated KU50 [132].

This ambitious endeavor would entail leveraging molecular marker technologies to discern the haplotypes present in KU50 from R1 and R90, coupled with accelerated inbreeding via induced flowering methods [133,134,135]. The ultimate aim would be to develop two (partial) inbred lines capable of generating hybrids closely resembling KU50 when crossed. While acknowledging the ambitious nature of this undertaking, once these two sets of partial inbred lines are established, cassava breeding could transition from the current phenotypic recurrent selection approach to a more dependable breeding system akin to that employed in maize, which has resulted in consistent genetic gains for so long. These partial inbred lines would parallel the venerable Mo17 and B73 inbred lines in maize, which have served as the foundation of commercial maize hybrids over the past four decades [129].

### 8.2. The Advantage of (Partially) Inbred over Non-Inbred Progenitors

In the absence of established heterotic patterns, it is still feasible to develop (partial) inbred lines from elite Thai germplasm and assess them in test crosses. However, early attempts to produce partially inbred cassava lines were halted due to the unavailability of methods for inducing early and profuse flowering. Despite reaching the S_3_ level of inbreeding, the process inadvertently favored the production of early branching genotypes, which were deemed unacceptable from an agronomic standpoint, though still vigorous enough to serve as progenitors in crosses. Notably, the first cassava genome sequencing utilized one of these S_3_ lines [8].

The induction of early and profuse flowering [133,134,135] would significantly accelerate and facilitate the production of inbred lines that do not exhibit early branching. Moreover, the pruning protocol [134] demonstrates enhanced efficacy (e.g., inflorescences with a larger number of flowers) in late branching materials. Eventually, doubled-haploid protocols may be developed, or a homologous version of the haploid inducer found in maize might be identified in cassava. In one way or another, the production of (partial) inbred lines in cassava is now indeed perfectly feasible.

Establishing a comprehensive database of breeding lines and parents can aid in estimating combining abilities and grouping clones into heterotic categories. Additionally, partially inbred lines can unveil latent desirable recessive traits, like waxy starch. Importantly, selected (partially) inbred lines inherently possess a reduced genetic load compared to their non-inbred progenitors. This confers notable advantages since, in theory, gametes from these lines should offer enhanced breeding value relative to the gamete pool produced by the original, non-inbred elite Thai varieties [23]. Moreover, by default, the SDGBV would be considerably smaller than in fully heterozygous progenitors, facilitating the identification and exploitation of breeding value.

Perhaps the most important advantage of using (partial) inbred lines as progenitors in future cassava breeding is the facilitation of trait introgression. The laborious and costly introgression of the waxy cassava starch trait into Thai elite germplasm marked one of the earliest documented cases of monogenic trait introgression in cassava. The ongoing efforts to introgress *CMD2* would be another significant effort. These experiences underscore the formidable challenges and expenses associated with trait introgression in cassava, especially when compared to crops with breeding programs based on homozygous progenitors.

The back-crossing scheme, probably the most employed widely breeding method, is impractical in cassava due to the absence of inbred progenitors. Interestingly, one of the most common applications of Marker-Assisted Selection (MAS) is in accelerated back-crossing [136,137]. The negligible use of molecular markers in cassava for MAS [138,139,140], therefore, is largely due to the impossibility of trait introgression through back-crossing.

### 8.3. Marker Assisted Breeding in Cassava

As next-generation sequencing technology advances, single nucleotide polymorphisms (SNPs) have emerged as the primary DNA marker. These SNPs are developed from biparental QTL analysis or genome-wide association studies (GWAS), replacing the previously popular simple sequence repeat (SSR) markers [141]. Marker-assisted recurrent selection and genomic selection have been instrumental in accelerating genetic gains by shortening selection cycles and increasing selection intensity, particularly in early cassava breeding generations. The high density of SNPs obtained through whole-genome sequencing has enhanced marker prediction accuracy, facilitating the discovery of genes associated with desirable traits for potential gene editing applications in the future.

Successful applications of marker-assisted selection in cassava breeding programs have primarily targeted traits with high heritability. For instance, Ige et al. (2021) [138] utilized Kompetitive Allele-Specific Polymerase Chain Reaction (KASP) assays for genotyping of three SNPs, two of which are linked to a major cassava mosaic disease (CMD) resistance locus on chromosome 12 (S12_7926132, S12_7926163), and one linked to a minor locus on chromosome 14 (S14_4626854), as reported by Rabbi et al. (2020) [107]. In IITA’s recurrent selection breeding, the overall predictive accuracy for CMD resistance in the breeding population reached 84%, whereas it was 71% in the pre-breeding population. The higher false-positive rate in the pre-breeding population was attributed to the lack of co-segregation between the favorable SNP allele and the resistant gene in progenitors from open-pollinated exotic cassava germplasm sourced from CIAT.

To effectively utilize these markers in Thai cassava breeding, high-density mapping through whole-genome sequencing is crucial for identifying true gene associations linked to CMD resistance. Similarly, for addressing other biotic stresses, like cassava brown streak disease (CBSD), biparental QTL mapping has revealed several loci associated with resistance traits. Ferguson et al. (2023) [142] identified candidate genes involved in the lignin pathway that play crucial roles in preventing viral replication in root mesophyll cells, particularly affecting leaf symptoms and root necrosis.

In addition to disease resistance, DNA markers have been pivotal in enhancing cassava for consumption purposes. GWAS analyses have identified loci associated with various root qualities, such as mealiness, fiber content, adhesiveness, taste, aroma, color, and firmness after boiling, shedding light on genes linked to carbohydrate metabolism, cell adhesion, secondary cell wall formation, and proteolytic activity during fruit ripening [143]. SNPs, such as those in the phytoene synthase gene, have been employed to select cassava lines with higher carotenoid content [144]. Furthermore, GWAS studies on starch paste properties in numerous cassava accessions have pinpointed SNPs associated with viscosity, pasting temperatures, and other key properties essential for specific starch industries [56,145].

For quantitative traits influenced significantly by environmental factors, GWAS analyses across diverse cassava accessions have unveiled numerous trait-associated loci, illustrating variability in loci identified across different studies. For example, Mbe et al. (2024) [146] identified 52 SNPs associated with traits related to nitrogen use efficiency and yield, highlighting traits such as stay-green ability, chlorophyll content, dry matter content, and yield components, like dry and fresh root yields. Variability in trait loci has also been observed under specific conditions, such as drought stress in Nigeria’s dry savannah, where distinct loci were identified for traits like fresh root yield, harvest index, and root number per plant [147].

In Thailand, the use of DNA markers for CMD and CBSD resistance is critical due to the importance of these traits for local cassava production. Preparation for potential CBSD outbreaks involves preemptive use of CBSD-resistant markers in breeding programs, given the disease’s absence in the region thus far. Introducing germplasm from Africa further supports efforts to sustain cassava production by enhancing resistance to CBSD and other emerging threats.

### 8.4. CRISPR/Cas9-Mediated Gene Editing in Cassava

Recent advances in CRISPR/Cas9 technologies have provided significant opportunities to enhance cassava’s agronomic traits, such as disease resistance and yield, promising substantial improvements in cassava’s yield and utility [148]. However, continued research, particularly in gene discovery, is crucial to realize the full potential of these biotechnological advancements. The CRISPR/Cas9 system involves delivering the Cas9 nuclease and a guide RNA into plant cells to induce DNA cleavages at specific genomic locations. The subsequent repair process, often via non-homologous end joining (NHEJ), introduces mutations that can knock out or alter gene function.

Initial applications of CRISPR/Cas9 in cassava demonstrated its capability for precise genomic editing by targeting the Phytoene desaturase (*MePDS*) gene, resulting in visible albino phenotypes in cotyledon-stage somatic embryos [149]. Following this experiment, the technique has been progressively applied to various traits, although initially limited to a few cassava varieties, such as TMS60444, known for their high efficiency in gene transformation. Despite these advancements, challenges remain in applying CRISPR technology to a broader range of cassava varieties due to limitations in the production of friable embryogenic calli (FECs) and transformation efficiency, which are influenced by genotype-specific transformation capabilities [150]. Nonetheless, recent studies on the cassava SC8 variety, primarily focused on the induction of somatic embryos and FECs, exploring genetic transformation have demonstrated opportunities to expand CRISPR technology to commercial varieties [151].

Applications of CRISPR/Cas9 in cassava have included enhancing disease resistance by targeting susceptibility genes for cassava mosaic disease (CMD) and cassava brown streak disease (CBSD), notably reducing disease severity and incidence. Editing multiple isoforms of eIF4E resulted in attenuated symptoms and decreased severity of CBSD root necrosis upon viral challenge [152]. However, generating cassava varieties resistant to cassava mosaic disease by expressing a CRISPR/Cas9 system targeting cassava mosaic virus has encountered challenges, such as the emergence of virus strains with single nucleotide mutations that evade CRISPR targeting [153].

Advances have also been achieved in reducing the presence of toxic cyanogenic glucosides by dual targeting the *CYP79D1* and *CYP79D2* genes, responsible for the initial steps in linamarin synthesis, thus reducing cyanogenic glucoside content in the edible parts of cassava [154,155]. Furthermore, CRISPR/Cas9 has facilitated modifications in starch bio-synthesis genes, such as *GBSS* and *PTST1*, altering amylose content and generating cassava varieties with tailor-made starch properties for diverse industrial applications [156]. Additionally, CRISPR editing of the starch branching enzyme 2 (*SBE2*) generated high-amylose starch cassava variants that exhibit enhanced nutritional qualities and industrial utility [157]. Moreover, CRISPR targeting of the feruloyl CoA 6′-hydroxylase genes (*MeF6′H1*, *MeF6′H2*, *MeF6′H3*), involved in scopoletin accumulation, has significantly delayed post-harvest physiological deterioration (PPD), extending cassava’s shelf life and reducing the economic losses associated with the rapid spoilage of cassava roots [158].

### 8.5. The Induction of Flowering

As discussed earlier, there is a rising interest in varieties with a non-branching plant architecture, also characterized by late or absent flowering. Branching and flowering are evidently governed by linked genetic factors and remain stable traits, although notably influenced by environmental conditions, like photoperiod duration and temperatures. Extended daylight hours and cooler temperatures tend to induce earlier flowering in genotypes that would otherwise flower late or not at all [133,159,160].

Recent achievements in inducing early and prolific flowering in cassava are poised to revolutionize its breeding practices. Studies have demonstrated that extended photoperiods can prompt earlier flowering in late-flowering germplasm. Furthermore, pruning techniques and the application of plant growth regulators have proven effective in stimulating abundant flowering [133,134,135,161]. The development and adoption of these flowering-inducing technologies have already substantially reduced the duration of crossing nurseries in the 2020s [133,134,135,161].

The pruning technique always results in the production of large inflorescences with large number of flowers. However, it requires that flowering/branching has already been induced. In high altitude environments (e.g., 1000 m above sea level) the extension of photoperiod (illumination with red lights during the night) had a clear impact shortening the time required for the first flowering event, which is the one where pruning of the young branches is typically done with best results. In Thailand, however, the extension of photoperiod may not be as efficient, except in high altitude research stations in Chiang Mai province in northern Thailand. However, this region is far away from the traditional location of crossing nurseries in the country. Further research is needed to reduce the time required for the first flowering event in non-branching or late-flowering genotypes, which are the most desirable because of their desirable plant architecture.

### 8.6. Improved Phenotyping Tools

For the above-ground growth estimation, simple ratio vegetation index and chlorophyll vegetation index were suggested for determining root yield potential (R^2^ of 0.50 and 0.57, respectively), whereas the plant height and canopy area provided more accurate predictions of root yield (R^2^ of 0.87 and 0.65, respectively) [162]. The application of the red–green–blue (RGB) and multispectral images could help in estimating plant growth during vegetative stage from 1–3 MAP with high accuracy for predicting biomass, chlorophyll content, net photosynthesis and leaf area [163], which can be used for high-throughput screening of cassava lines in breeding for abiotic stress tolerance. The application of aerial-image based phenotyping coupled with a machine-learning model was applied using ground-truth data together with multispectral data for vegetative indices. The key stages that the multiple linear regression model could estimate both above-ground and underground biomass were the elongation and late bulking stages [164].

Starch content in cassava was evaluated by the near-infrared (NIR) spectroscopy sensor that correlated with the percentage of the content of dry starch extracted from root to the fresh root weight with R^2^ of 0.84–0.90 [165]. Near-infrared spectroscopy has been employed for identifying waxy genotypes from the seeds. However, specificity values of the prediction ranged between 0.27 and 0.74 for selection of seeds for waxy genotypes [166]. As for the quality of cassava product for consumption, amylose and amylopectin content, starch granule structure, pasting properties, and fresh and boiled surface properties could be phenotyped by near-infrared spectroscopy and hyperspectral sensors [167], along with dry matter content and water absorption, to determine the cooking quality of cassava [168]. Carotenoid content is routinely predicted for fresh root tissue using desk or portable NIRs [167].

Cassava root diameter distribution was evaluated in KU50 weekly during 3 to 12 weeks of growth in the field. Distribution of root diameters can be imaged and assessed through a video acquisition box [169]. Moreover, 3D-root images from 11 months old plants grown in the field have been used to predict crown volumes, crown surface, root density, surface-to-volume ratio, root numbers, root angle, crown diameter, cylinder soil volume, CRC compactness and root length [170]. This protocol, used for the study of roots from 130 partially inbred lines and 6 parents, linked SNPs with root weight and other relevant root traits [171]. Oliveira et al. [172] reported that the number of roots was the only factor having a direct effect on root weight. The digital image of the root traits platform (DIRT) [173] was utilized for top view and side view images of R5, R9, R11, HB60 and KU50 at 3-, 10- and 12-month-old harvest for determining the drought effect on cassava yield and showed that the root yield under drought condition was reduced by 80%, and DIRT assessed root yield with high correlation with the manual root weight method, with R^2^ of 0.86 [174].

For non-destructive root growth evaluation, cassava root architecture in the pot was assessed by computed X-ray tomography (CTX) with the automated root segmentation approach (Rootforce) [175]. Later, CTX was combined with the growth chamber named chamber #8 as controlled environment for root development investigation, which can be integrated with other sensors, such as hyperspectral or near-infrared sensors, in the future [176]. Moreover, Alle et al. [177] investigated the application of combining deep convolutional neural networks (DCNNs) with a weakly supervised learning paradigm and a spatial pyramid pooling (SPP) layer together with a specialized iterative inference algorithm, providing better segmentation of root structure than an analytical reference algorithm for root segmentation. However, the CTX application is limited to plants grown in pots withing a growth chamber. Thus, this methodology might not be suitable for breeding purposes. The application of ground-penetrating radar (GPR) was demonstrated to estimate root length and width in the different angles in the field without root evacuation [178]. The linear regression model to estimate root yield of different cassava varieties had a relatively high regression coefficient ranging from R^2^ of 0.51 to 0.77 [179]. Agbona et al. [180] tried to use the GPR for measuring early storage root bulking. However, the different environmental conditions in the field, such as temperature, humidity and solar radiation, were required for more accurate predictions of root yield. Moreover, the GPR was recommended not only for selecting root yield in the field, but also to help normalization of the spatial heterogeneity of the yield trials [181].

## 9. Conclusions

The history of cassava breeding in Thailand highlights the critical role of collaboration with the International Center for Tropical Agriculture (CIAT) in providing valuable international cassava germplasm for Thai variety development. The rapid growth of cassava production in Thailand and Southeast Asia, driven by global demand for animal feed and starch-based industries, underscores the need for high-yield varietal development. The significant influence of genotype-by-environment interactions and non-additive genetic effects on fresh root yield emphasizes the importance of institutional collaboration between government research bodies and private processing industries. This partnership can facilitate multi-location trials and the use of rapid propagation technologies, expediting the breeding program. The success of the KU50 variety and its progenies demonstrates the efficacy of recurrent selection, despite the lengthy phenotypic selection cycles. Incorporating genomic selection in early yield trials could accelerate recurrent selection for high-heritability traits, such as dry matter content. Marker-assisted selection (MAS) is currently used for tracking CMD2 resistance to cassava mosaic disease and will be applied for CBSD resistance screening, preparing for future biotic stresses in Thailand. With the increasing occurrence of climate extremes, the impact of abiotic and biotic factors on crop productivity is a growing concern. Breeding objectives must incorporate traits related to climate resilience and disease resistance to sustain and maximize yield potential. Conventional breeding methods have struggled to facilitate the backcrossing process required to incorporate these traits into elite varieties. Gene editing offers potential, but developing efficient tissue regeneration methods for each variety remains challenging. Implementing recurrent selection with partially inbred progenitors exhibiting desirable combining ability is a promising strategy for increasing cassava yield and introgressing desirable traits into partially inbred lines. Achieving these goals requires a partnership between molecular research and cassava breeding programs.

## Figures and Tables

**Figure 1 plants-13-01899-f001:**
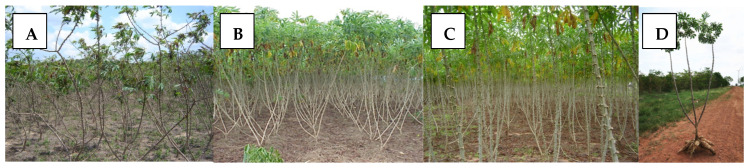
Branching types of cassava. (**A**). Branching variety KU50; (**B**–**D**) are non-branching varieties HB60, HB80 and HB100, respectively.

**Figure 2 plants-13-01899-f002:**
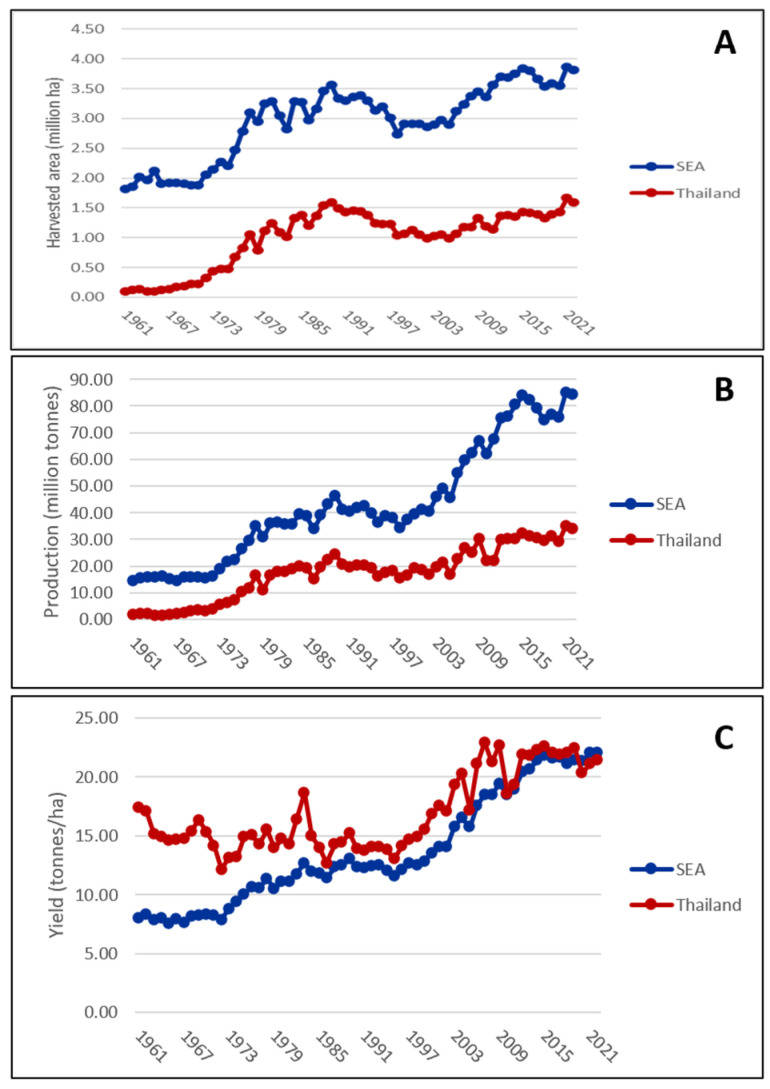
Annual data for cassava harvested area (**A**), production (**B**) and yield (**C**) in Thailand and Southeast Asia Countries [11].

**Figure 3 plants-13-01899-f003:**
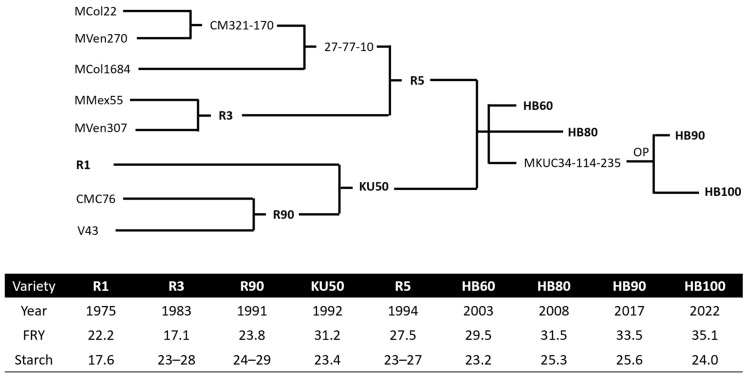
The pedigree of Huaybong cassava varieties, HB60, HB80, HB90 and HB100, released by Thai Tapioca Development Institute (TTDI) and Kasetsart University (KU) from the breeding program in the Department of Agronomy, Faculty of Agriculture, KU. Year: releasing year; FRY: fresh root yield in t/ha; Starch: starch content (percentage). Diagram was modified from [23]. Root yield and starch content information were retrieved from [21] for R3, R90, R5 and from [64] for R1, KU50, HB60, HB80, HB90 and HB100 in 74 yield trial locations in 11 provinces in Thailand from 2012 to 2021.

**Table 1 plants-13-01899-t001:** Overall performance of elite Thai varieties that have been released through the 1990s. Data summarizes agronomic results in farm trials conducted in 53 locations [23,62].

Cultivar	Fresh Root Yield	Dry Matter Content	Dry Root Yield	Regression
	(t/ha)	(%)	(t/ha)	Coefficient (b) for FRY
Rayong 1	22.4	30.7	6.9	0.84
Rayong 3	19.0	34.2	6.5	1.02
Rayong 60	23.6	31.5	7.4	1.07
Rayong 90	23.5	34.7	8.2	0.98
KU 50	24.8	34.0	8.4	0.96

**Table 2 plants-13-01899-t002:** Overall performance of four elite Thai varieties in farm trials conducted in 11 locations from 1999 to 2000 in Thailand [69].

Cultivar	Fresh Root Yield	Starch Content	Dry Root Yield
	(t/ha)/Regression Coefficient (b)	(%)/Regression Coefficient (b)	(t/ha)/Regression Coefficient (b)
KU50	36.25/1.39	25.61/1.01	13.44/1.01
HB60	36.25/0.93	25.55/0.89	13.06/0.74
Rayong 5	35.13/1.04	24.70/0.74	12.81/1.49
Rayong 72	31.31/0.65	22.88/1.36	11.75/0.76

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
