# Peer review of "Cassava Breeding and Cultivation Challenges in Thailand: Past, Present, and Future Perspectives"

_plants, 2024, doi:10.3390/plants13141899_

Round 1

Reviewer 1 Report

Comments and Suggestions for Authors

Dear Author,

Please find the comments in the manuscript attached. 

Author Response

Dear Reviewer 1,

We are submitting a revised manuscript Plants-3055657 entitled “Cassava Breeding and Cultivation Challenges in Thailand: Past, Present, and Future Perspectives” We would greatly appreciate it if you would consider this manuscript for publication in your journal.

In our revision based on the reviewers’ suggestions, we have highlighted the changes in yellow. The green highlights indicate retained information from the previous version, which we used to respond to the reviewer’s comments and intend to keep in the revised manuscript. The comment boxes show the parts deleted as per the reviewer's suggestion.

Please find the attached file here for the responses to your suggestions.

We appreciate the constructive feedback, which has helped us improve the clarity and comprehensiveness of our manuscript.

Best regards,

Corresponding author of Plants-3055657

In the case of attachment could not be opened please find this following responses

Comment 1: mentioned the recent year data for productions 

Response to comment 1: Please see line 59-67 in the revised version of manuscript

Comment 2: Mentioned the year of start the programm 

Resonse to comment 2: According to the document, it was in the early 1980s. We have added in the line 69 in the revised version.

Comment 3: re-evaluated in year ??

Response to comment 3: We have rewritten line 97-102 to clarify this. The re-evaluation were performed in many countries importing germplasm from CIAT and SNP study was conducted as citation [18] during 2016-2021 in CIAT germplasm to reduce redundancy.

Comment 4: Role of markers assisted breeding and QTL mapping need subheading and more explainations. 

Response to comment 4: Thank you for your valuable suggestion, we have added the new section in 8.3. We put it in the future direction instead of the history of cassava breeding because, in Thailand, we do not include MAS regularly in our cassava breeding due to the limit of funding. However, in the 8.3 section Line 858-911, we hope it will show the significant benefit of MAS for the breeding enough to persuade our funding agency policy makers to understand the point.

Comment 5: Add the value of production 

Response to comment 5: (14.9 t/ha) in line 410 was averaged from data from FAOstat from 1975-1982 for the average cassava yield in Thailand. For R1 yield, please see Table S2.

Comment 6: Genome editing is one of the most promissing area for the crop improvements therefore, author must add one heading and subheading of genome editing and the role of genome editing in the cassava improvements 

Response to comment 6: There was the section 8.4 Line 913-954 highlight the studies of gene editing in cassava improvement.

Comment 7: Redice the conclusion and rewrite major points of conclusion. 

Response to comment 7: We have revised the conclusion higlighting the major key of success in cassava breeding in Thailand and the suggestion for the future direction.

Reviewer 2 Report

Comments and Suggestions for Authors

It is better to add a table of contents at the beginning, which is beneficial for readers to read.

Author Response

Dear Reviewer 2,

We are submitting a revised manuscript Plants-3055657 entitled “Cassava Breeding and Cultivation Challenges in Thailand: Past, Present, and Future Perspectives” We would greatly appreciate it if you would consider this manuscript for publication in your journal.

In our revision based on the reviewers’ suggestions, we have highlighted the changes in yellow. The green highlights indicate retained information from the previous version, which we used to respond to the reviewer’s comments and intend to keep in the revised manuscript. The comment boxes show the parts deleted as per the reviewer's suggestion. We appreciate the constructive feedback, which has helped us improve the clarity and comprehensiveness of our manuscript.

For your comment of table content, we are not sure if it is fit with the journal format so that we have added the brief list of section at the end of the introduction in line 74-78 as "

In this review, there are the following sections: 2. Global Cassava Genetic Resources; 3. Special Features of Cassava Physiology, Reproductive Biology, Roots and Starch; 4. Cassava Breeding in Thailand; 5. Released Commercial Cassava Varieties in Thailand; 6. The impact of KU50 in Thailand and Southeast Asia; 7. Determinants of Cassava Yield and Production Quality; 8. Future of Cassava Breeding Direction".

Best regards,

Corresponding author of Plants-3055657

Reviewer 3 Report

Comments and Suggestions for Authors

The authors present their study on the cassava breeding and cultivation challenges in Thailand. In this manuscript, the authors present an examination of the practice and historical development of cassava breeding in Thailand. These results provide a basis for further studies on the cassava breeding program in order to achieve their goals and this give some useful information about cassava breeding and industrialization to other regions. The manuscript was well written with all results available for reference. I recommend acceptance with minor revisions.

1.       Add the details of Manihot species in a table form, especially the endemic species.

2.       Global cassava genetic resources should include China. How about China's role in genetic resources collection?

3.       Abstract: Consider adding a brief mention of the methodologies used in historical analysis and future breeding techniques. Expand on the specific challenges posed by climate change and how the future directions aim to address these.

4.       Line 44 is not suitable here. The previous sentence talks about lineage, and the next sentence also talks about lineage. Introducing the sweet and bitter types in between is out of place.

5.       Line 47-53: Provide more context on the significance of Mexico and Central America as biodiversity hubs for Manihot species.

6.       Line 54: This section should include more details about cassava cultivation in Thailand, including current cultivation area, regions, and yield.

7.       Line 63-125: Please highlight the contributions of CIAT's genebank to global cassava research and breeding efforts.

8.       Lines 74-76: The total does not add up to 100%. Where is the missing 6%?

9.       Line 116: Since 1970, what is the current status of Thailand's genetic resources, including the introduction of 100,000 seeds? Have any new genetic resources been domesticated?

10.   Line 126-191: Please provide examples of how this flexibility impacts cassava’s market value and farmers’ incomes.

11.   Line 127-191: Please provide examples of how this flexibility impacts cassava’s market value and farmers’ incomes.

12.   Line 138: Why should the stems be bundled together in groups of 50 stems each?

13.   Line 139: Can the length really vary from 1 to 2 meters? Please confirm, as in production, stems typically have 5 buds and are usually over 15 cm long.

14.   Line 240: This sentence has already been mentioned earlier. This section should introduce the breeding process, such as how many hybrid combinations could produce how many variations, and the general rules.

15.   Line 343-380: Please describe the performance and promotion of these high-carotenoid content, excellent cooking quality, and disease-resistant varieties in actual agricultural production.

16.   In addition, could you present the mechanisms behind PPD in more detail, including the role of reactive oxygen species (ROS)? Could you provide a summary of the key achievements and successful varieties developed through the CIAT/DOA collaborative program?

Comments on the Quality of English Language

no comments

Author Response

Dear Reviewer 3,

We are submitting a revised manuscript Plants-3055657 entitled “Cassava Breeding and Cultivation Challenges in Thailand: Past, Present, and Future Perspectives” We would greatly appreciate it if you would consider this manuscript for publication in your journal.

In our revision based on the reviewers’ suggestions, we have highlighted the changes in yellow. The green highlights indicate retained information from the previous version, which we used to respond to the reviewer’s comments and intend to keep in the revised manuscript. The comment boxes show the parts deleted as per the reviewer's suggestion. We appreciate the constructive feedback, which has helped us improve the clarity and comprehensiveness of our manuscript.

Please find the attachment for the responses to your suggestion.

Best regards,

Corresponding author of Plants-3055657

Attached responses 

Thank you for your feedback, which has helped us enhance the clarity and comprehensiveness of our manuscript.

  1. Add the details of Manihot species in a table form, especially the endemic species.

Response from authors: We added supplementary table S1 for wild Manihot species as mentioned in Line 43-45.

  1. Global cassava genetic resources should include China. How about China's role in genetic resources collection?

Response from authors: We added one paragraph about genetic resource and breeding program from China in Line 146-154.

  1. Abstract: Consider adding a brief mention of the methodologies used in historical analysis and future breeding techniques. Expand on the specific challenges posed by climate change and how the future directions aim to address these.

Response from authors: We have revised the abstract highlighting more methodologies and add the key success and try the best to expand the specific challenges as your recommendation and to fit with 200-word limitation.

  1. Line 44 is not suitable here. The previous sentence talks about lineage, and the next sentence also talks about lineage. Introducing the sweet and bitter types in between is out of place.

Response from authors: The sentence “There are two distinctive types of cassava, sweet and bitter, depending on the levels of cyanogenic glucosides.’ In Line 48 was deleted.

  1. Line 47-53: Provide more context on the significance of Mexico and Central America as biodiversity hubs for Manihot species.

Response from authors: Line 39-41 “There were two hubs of biodiversity for Manihot species which were Brazil for 80 species and Central America for the rest” was added.

  1. Line 54: This section should include more details about cassava cultivation in Thailand, including current cultivation area, regions, and yield.

Response from authors: Line 59-64, the details about cassava cultivation in Thailand, including current cultivation area, regions, and yield were added.

  1. Line 63-125: Please highlight the contributions of CIAT's genebank to global cassava research and breeding efforts.

Response from authors: There were details in Line 92-93 (in green) and a paragraph from Line 132-145 indicating the contributions of CIAT's genebank to global cassava research and breeding efforts (in both green and yellow).

  1. Lines 74-76: The total does not add up to 100%. Where is the missing 6%?

Response from authors: The numbers shown in the website were round digits which were stated as “approximately percentage” without decimal point were added up to 96% (37+24+21+7+7). The missing 4% could be the decimal points which were not shown in the original source of information.

  1. Line 116: Since 1970, what is the current status of Thailand's genetic resources, including the introduction of 100,000 seeds? Have any new genetic resources been domesticated?

Response from authors: The detail in line 134-138 was added.

  1. Line 126-191: Please provide examples of how this flexibility impacts cassava’s market value and farmers’ incomes.

Response from authors: Line 156-159 was added and Line 162-164 was added to explain how this flexibility impacts cassava’s market value and farmers’ incomes.

  1. Line 127-191: Please provide examples of how this flexibility impacts cassava’s market value and farmers’ incomes.

The same response as 10.

  1. Line 138: Why should the stems be bundled together in groups of 50 stems each?

Response from authors: Line 172-173: “to be stored in an upright position in the shade and on the soil with regularly watering” was added.

  1. Line 139: Can the length really vary from 1 to 2 meters? Please confirm, as in production, stems typically have 5 buds and are usually over 15 cm long.

Response from authors: Line 174-176: “The length of these stored stems can vary from 1 to 2 meters depending on the cultivars having branching or non-branching plant type and growing conditions. The stem will be cut into 20-centimeter-long stakes with 5 to 7 nodes immediately before cultivation.” Was added.

  1. Line 240: This sentence has already been mentioned earlier. This section should introduce the breeding process, such as how many hybrid combinations could produce how many variations, and the general rules.

Response from authors: Line 253-261 were added. Moreover, Line 284-286: “A large amount of F1 hybrid seeds should be obtained from several crosses as the ex-ample from CIAT cassava breeding program mentioned earlier.” was added.

  1. Line 343-380: Please describe the performance and promotion of these high-carotenoid content, excellent cooking quality, and disease-resistant varieties in actual agricultural production.

Response from authors: Line 423-432-419: A paragraph was added.

  1. In addition, could you present the mechanisms behind PPD in more detail, including the role of reactive oxygen species (ROS)?

Response from authors: Line 218-219: “resulting in the oxidation of polyphenolics and their glucosides which are mostly scopoletin and scopoline” was added to explain the ROS role.

Could you provide a summary of the key achievements and successful varieties developed through the CIAT/DOA collaborative program?

Response from authors: Line 389-398 have explained the key achievements and successful varieties developed through the CIAT/DOA collaborative program.

We hope these responses cover all your supportive suggestions.

Best regards,

Corresponding Authors of Plants-3055657

Round 2

Reviewer 1 Report

Comments and Suggestions for Authors

NA